# Recent Advances in the Synthesis of Polyamine Derivatives and Their Applications

**DOI:** 10.3390/molecules26216579

**Published:** 2021-10-30

**Authors:** Artemiy Nichugovskiy, Gian Cesare Tron, Mikhail Maslov

**Affiliations:** 1Lomonosov Institute of Fine Chemical Technologies, MIREA—Russian Technological University, 86 Vernadsky Ave., 119571 Moscow, Russia; nichugovskij@mirea.ru; 2Dipartimento di Scienza del Farmaco, Università del Piemonte Orientale, Largo Donegani 2, 28100 Novara, Italy; giancesare.tron@uniupo.it

**Keywords:** polyamines, polyamine conjugates, antitumor activity, antimalarial agents, Ugi reaction

## Abstract

Biogenic polyamines (PAs) are involved in the growth and development of normal cells, and their intracellular concentration is stable. The concentration of PAs in cancer cells is significantly increased to promote and sustain their rapid proliferation. Over the years, synthetic PAs, which differ in their structure, have demonstrated high antitumor activity and are involved in clinical trials. The chemical synthesis of PAs and their conjugates require the correct choice of synthetic pathways—methods for constructing conjugates and the orthogonal protection of amino groups. The most common methods of synthesis of PA conjugates are acylation of regioselectively protected PAs or their alkylation under the conditions of the Fukuyama reaction. One of the most promising methods of PA synthesis is the use of a multicomponent Ugi reaction, which allows various PAs to be obtained in high yields. In this review, we describe and analyze various approaches that are used in the synthesis of polyamines and their conjugates.

## 1. Introduction

Natural polyamines (PAs), as shown in Figure 1, are low-molecular-weight, positively charged molecules at the physiological pH. They are found in all eukaryotic cells in millimolar concentrations [1] and are involved in many physiological cellular functions, such as cell growth, proliferation, migration, differentiation, immunity, gene regulation, and DNA stability. In addition, they are indirectly involved in the synthesis of proteins and nucleic acids [2,3]. They can form complex structures with DNA, RNA, ATP, and certain types of proteins and phospholipids [4]. Fully protonated PAs cannot pass through the cell membrane by passive diffusion. In mammalian cells, exogenous PAs need active transport to cross the plasma membrane. Once inside the cell, they distribute in all cellular components, owing to their high solubility.

In highly proliferative cells, the concentration of PAs increases sharply. In many types of cancer, the intracellular concentration of PAs was found to be 4–6 times higher than in corresponding normal tissues [5]. The concentration of PAs in a normal cell is maintained within a certain range. When its level is reduced, cell proliferation and migration stop. On the other hand, an excess of PAs leads to programmed cell death—apoptosis [6,7].

Of note, the lack of PAs in cells leads to the activation of apoptosis; therefore, a promising anticancer target is the inhibition of PA synthesis [8,9]. Additionally, Chen’s work [10] describes the mechanisms of trapping natural PAs in cells because of their binding to peptide-pillar[5]arene conjugates, which show high antitumor activity against breast cancer.

The complete metabolism of PAs involves seven enzymatic reactions (Figure 2). In mammalian cells, the first step is the conversion of arginine to ornithine by the enzyme arginase. Ornithine can also enter the cell from the circulating blood plasma, where it is decarboxylated by ornithine decarboxylase (ODC) to form putrescine. Putrescine is converted to spermidine by aminopropyl transferase and spermidine synthase. Spermidine is further converted to spermine. Methionine adenosyltransferase transforms methionine to *S*-adenosylmethionine (AdoMet), which is then decarboxylated by *S*-adenosylmethionine decarboxylase (AdoMetDC) to form the aminopropyl intermediate dcAdoMet. The side-product in this reaction—5′-methylthioadenosine—is converted back to adenosine.

The number of PAs in the cell is carefully controlled by a combination of three key enzymes: ODC, AdoMetDC, and *N*^1^-acetyltransferase (SSAT). PA biosynthesis is regulated by ODC and AdoMetDC, and the catabolism is regulated by *N*^1^-acetyltransferase (SSAT). The activity of biosynthetic enzymes is low in non-proliferating cells, but it increases in response to various growth stimuli [5]. Potential cancer chemoprevention strategies include the inhibition of PA synthesis or induction of PA catabolism. An increased PA concentration is associated with a high rate of cell proliferation, a decrease in apoptosis, and an increase in the expression of genes that affect tumor invasion and metastasis [11].

PAs can be trapped from exogenous sources. The polyamine transport system (PTS) is known to involve the absorption and withdrawal of PAs, which are necessary for their processing. The molecular structure of the mammalian PTS still remains, but the most preferred PA transport is linked to glypican-mediated endocytosis, pinocytosis, and caveolin-mediated endocytosis [12,13]. PTS is not highly specific and can transfer other molecules (Figure 3), such as paraquat, methylglyoxal bis(guanylhydrazon) (MGBG), mepacrine, and polyamine-based compounds. The amount of transported compound increases when the delivering molecule resembles the structure of spermidine or spermine. Additionally, the data in the literature indicate that the primary amino groups in the PA structure are not sufficient for their uptake [14]. PTS can be considered a target for cytotoxic polyamine-conjugated drugs [15].

In this regard, PAs are potential influencers on cellular processes. Some of their synthetic analogs have already found applications in preclinical and clinical trials [16,17,18,19,20,21,22].

Several synthetic approaches to obtaining analogs and conjugates of PAs are known. The optimal synthetic strategy depends on the structure of the desired molecules. In this review, we describe and analyze various approaches that have been used in the synthesis of polyamines and their conjugates.

## 2. Methods for Polyamines Synthesis

### 2.1. Alkylation

The simplest way to obtain amine derivatives that include PAs is via their *N*-alkylation with haloalkanes, which usually leads to the formation of a mixture of secondary amines **2**, tertiary amines **3**, or quaternary ammonium salts **4** (Figure 1) [23].

In the synthesis of anti-tuberculosis agents based on pentacycloundecane tetramines, an excess of starting cyclic diamines (piperazine or homopiperazine) was successfully used to obtain the desired molecule **8** (Figure 2) [24,25]. Alkylation with crotyl bromide **5** gave compound **6**, which was refluxed with the pre-synthesized tosylate **7** to form tetramine **8** with different chain lengths in high yields.

The alkylation of amines was also used in the synthesis of PA–naphthalimide conjugates **12a**,**b** (Figure 3). Bromides **10a**,**b** reacted with di-Boc-protected homospermidine **9**, and the unstable intermediate formed with a free amino group required complicated purification. In this regard, the fully protected compounds **11a**,**b** were prepared in 40% yield in two steps. After Boc removal under acidic conditions, compounds **12a**,**b** were obtained as hydrochloride salts in 60% yield [26].

Another method used for grafting an amino group into a substrate molecule is the Gabriel reaction, where phthalimide is initially reacted with alkyl halides followed by the cleavage of the intermediate. In this case, the interaction of 3-chloro-2-chloromethyl-1-propene **13** and potassium phthalimide gave a mono alkylated adduct **14**, which then reacted with *o*Ns-protected putrescine (Figure 4). After the Boc-protection of the secondary amino group **15**, phthalimide was removed by hydrazine to form tetramine **17** [27].

### 2.2. Acylation

A conversion of amines to amides by *N*-acylation is one of the fundamental, reversible organic reactions that are catalyzed either by acids or bases [28]. In the synthesis of a PA-benzotriazole derivative **21a**–**f** (Figure 5), an amide linker was formed by the treatment of carboxylic acid **18** with amine **19a**–**f** in the presence of BOP [29].

Biocatalysts make obtaining enantiomerically pure compounds in high yields possible. The stereoselective synthesis of PA precursors from non-activated esters and α,ω-diamines catalyzed by *Candida antarctica* lipase (CAL) in organic solvent gave bis-(amidoesters) as the only product [30]. Further conversion of enzymatically obtained precursors into PAs was carried out according to the simple two-stage protocol: ammonolysis and subsequent reduction.

CAL catalyzes the enantioselective acylation of racemic diamines, leading to optically active bis-(amidoesters) with a moderate to excellent enantiomeric excess (ee). In the case of *rac*-1,2-diaminopropane and dimethylmalonate **22**, the resulting ee of (*R*)-(+)-bis-(amidoester) **23** was more than 95% (Figure 6). The subsequent reduction of amide groups yielded optically active PA **24** [30].

### 2.3. Imine Formation

One method for the synthesis of PAs utilizes the Schiff bases, followed by their reduction [23].

The action of excess 2-butyne-1,4-diamine (**25**) on 3-azidopropanal leads to iminoazide **26**, which can be reduced by sodium cyanoborohydride to form azido alkyne **27**. The reduction of compound **26** with lithium aluminum hydride gives triaminoalkene **28**, whereas its reduction by triphenylphosphine leads to tri-amino alkyne **29** (Figure 7) [31].

The Golding method is also for the synthesis of secondary amines and uses a dibenzyltriazone protecting group (DBT) (Figure 8). The azide **30** is first converted to the corresponding iminophosphorane, followed by treatment with the aldehyde. The resulting product is reduced to form a protected spermidine derivative **31**. The removal of the DBT group is carried out in an aqueous solution of piperidine to give spermidine [23].

### 2.4. Michael Reaction

The addition reaction of amines to a Michael acceptor such as acrylonitrile is a frequently used method for the synthesis of PAs. Thus, the Michael reaction with the participation of primary amine **32** and acrylonitrile led to mixtures of mono- (**33**) and bis-cyanoethylated (**34**) amines, which could be reduced to linear or branched PAs (Figure 9) [23]. Then, the secondary nitrogen of the linear diamine **33** can be protected with methyl chloroformate to yield compound **35**. The cyanide group is then reduced by hydrogen on Raney-Ni followed by cyclization to give compound **36**.

### 2.5. Mitsunobu Reaction

The Mitsunobu reaction is a useful way to synthesize Pas from alcohols, diethyl azodicarboxylate (DEAD), triphenylphosphine (Ph_3_P), and sulfonamides. The reaction, which occurs under mild conditions, is often used in the synthesis of natural and other complex compounds. In [32], the reaction between ©-3-(*N*-methyltrifluoromathenesulfonamido)butan-1-ol **37** and 1,7-bis-(trifluoromethanesulfonamido)heptane **38** in the presence of DEAD and triphenylphosphine resulted in compound **39**, which was further subjected to protection/deprotection procedures to form optically active tetramine **40** (Figure 10). *N*,*N*,*N*′,*N*′-tetramethylazodicarboxamide (TMAD) or 1,1′-(azodicarbonyl)dipiperidine (ADDP) can also be used as an activating agent in a Mitsunobu reaction.

### 2.6. Fukuyama Amine Synthesis

Fukuyama amine synthesis provides powerful access to secondary amines **49** under mild reaction conditions. Nitrobenzenesulfonamide **48** (nosyl (Ns) amide) is acidic enough to be alkylated under Mitsunobu conditions, which is possible using weak bases and alkyl halides.

This method was demonstrated in [33], where spermine was converted to a per-nosyl derivative **41** with subsequent modification of the terminal nitrogen atom by compound **42** to form an asymmetric derivative **43**. Further cross-linking was performed using 1,4-ditosyloxybutane to form a symmetric product **44**. The subsequent removal of nosyl protecting groups was carried out in the presence of PhSH to form the desired product **45**. Refluxing **45** in TFA gave dimeric PA with 16 amino groups (Figure 11).

### 2.7. Solid-Phase Synthesis

In contrast to synthesis in solution, solid-phase organic synthesis significantly simplifies the stages of isolation and purification of the target PAs. In a work that described the synthesis of PAs in the solid phase, propanediamine was first attached to resin to form *N*-trityl-propanediamine **46** (Figure 12) [34]. Further chain extension was carried out using the activated ester method using the treatment of amine **46** with acids **47a**,**b** and **49** in the presence of HBTU and HOBt. The reduction of amides **50a**,**b** was achieved under mild conditions with borane to give tetramines **51a**,**b**. Finally, resin removal in the presence of TFA and tris-(trimethylsilyl)silane (TTMSS) resulted in the desired PAs **52a**,**b**.

The 2-chlorotrityl resin is the most recent resin modification used for solid-phase synthesis. The presence of the electron-withdrawing group improves its acidic stability and broadens its applications for acid-sensitive reactions. For a comparison, Table 1 shows the stability of trityl and 2-chlorotrityl esters of 4-hydroxypentanoic acid [35]. The application of 2-chlorotrityl resins was illustrated for the solid-phase synthesis of PAs [36,37,38].

For the synthesis of PA–naphthoquinone conjugates **55a**,**b**, HOBt-modified resin was used as an activating agent for *N*-nucleophiles (Figure 13). The synthesis of amides **55a**,**b** was carried out as a two-stage procedure: (i) the formation of a polymer-bound activated carboxylic acid ester **54** using PyBrop and naphthoquinone-carboxylic acid **53**, and (ii) the interaction with primary amines and cleavage of the amides. The polymer reagent acts as a leaving group; hence, simple filtration allows the desired compound **55a**,**b** to be easily isolated [39].

### 2.8. Regioselective Protection of Amino Groups

An optimal choice of protecting groups is necessary to create unsymmetric PA derivatives substituted at the terminal amino groups. Figure 14 describes an elegant method for the synthesis of tri-Boc spermine **58** with a free terminal amino group that can be further modified (e.g., the creation of Schiff base, acylation, or alkylation) [40]. Spermine reacts with a stoichiometric amount of 2-hydroxybenzaldehyde to give the Schiff’s base **57**. Boc protection, followed by the removal of the 2-hydroxybenzimide group by methoxyamine, gives tri-Boc-protected spermine **58** at 45% yield [41].

Another method of creating a free terminal amino group utilizes 0.5 eq. ethyl trifluoroacetate, which allows compound **58** to be obtained in a good yield over three steps (Figure 15) [42].

Furthermore, another technique for the regioselective protection of PAs involves spermine treatment with two-fold excess of ethyl trifluoroacetate to protect the primary amino groups (Figure 16) [43,44]. In this approach, secondary amino groups were protected by Boc_2_O to form compound **59**. The subsequent hydrolysis of trifluoroacetates with NaOH led to the formation of compound **60** with free terminal nitrogen atoms.

### 2.9. Cleavage of the Nitrogen-Containing Cycle

A method for obtaining PAs is the cleavage of a nitrogen-containing ring by the addition of an alkene or amine. Indium bromide [45], iron complexes [46], and scandium(III) triflate [47] can efficiently catalyze the aminolysis of aziridines **62** with aromatic amines **63** under mild conditions to form corresponding vicinal diamines **64** in high yields (Figure 17).

### 2.10. Opening of the Oxygen-Containing Cycle

Oxiranes are characterized by high reactivity under the action of nucleophilic agents such as hydroxide and alkoxide ions, amines, Grignard reagents, and hydride ions. In the first step, the oxygen atom of the oxirane ring is protonated to form the oxonium ion. Then, the three-membered ring opening under the action of *N*-nucleophile occurs strictly stereospecifically by an S_N_2 mechanism, accompanied by the inversion of the configuration of the asymmetric centrum [48]. The opening of the oxirane ring of compound **66** with the *N*-nucleophiles **65a**–**c** mainly occurs at the terminal carbon atom (Figure 18) [49,50].

## 3. Reduction

The reduction of nitrogen-containing functional groups is a key method for the preparation of PAs. Most reduction reactions are carried out in the presence of reagents that transfer hydrogen from aluminum, boron, and silicon compounds. Some of these reagents provide a significant degree of chemo- and stereoselectivity (Table 2).

A major advantage of LiAlH_4_ over other reductants is its ability to reduce almost all carbonyl groups, with the exception of carbamates (the protecting groups Cbz and Boc can be resistant to the reduction of LiAlH_4_). The yields are usually high, and the chiral centers of the carbonyl carbon remain non-racemized.

Sodium borohydride (NaBH_4_) is the most common reagent used to reduce aldehydes, ketones, and acid chlorides to the corresponding alcohols [58]. A reduction with borohydride proceeds selectively under mild conditions. Sodium borohydride is not suitable for the reduction of nitriles to amines but can be used to produce borane in situ using BF_3_·OEt_2_ or iodine [59]. The structural modification of one to three hydrogen atoms with different substituents leads to variations in the strength of the reducing agent. Substitution with electron-withdrawing groups, such as cyano and acyloxy, reduces the power of the reagent. For example, sodium cyanoborohydride (NaBH_3_CN), a much milder reducing agent, can only react with aldehydes and ketones under acidic conditions [60].

Boranes are very effective reducing reagents for amides and are somewhat more chemically selective than aluminum hydrides. Esters, alkyl halides, carbamates, epoxides, and nitro groups are not borane-reduced, making them ideal for amide reduction [61]. In addition, the initial product is an amino-boron adduct, which must be cleaved during processing. This is often not a problem for tertiary or aromatic amines, but primary and secondary aliphatic amines may require strong conditions for hydrolysis of the adduct, such as refluxing in 2M HCl.

Reduction by silanes is a chemoselective process because esters, epoxides, nitriles, and other sensitive functional groups are not affected. However, the reduction of primary amides is a challenge since nitrile dehydration is often observed. In this case, PhSiH_3_, TMDS [62], and Me_2_PhSiH [63] are usually used as hydrogen donors.

Hantzsch ester (HEH) is a readily available analog of the biological reductant nicotinamide adenine dinucleotide. This non-metallic hydrogen donor is more commonly used for the reduction of imines and electrophilic alkenes. The reduction with HEH has a high degree of chemoselectivity similar to that of silanes, but steric hindrance is a negative factor for this reducing agent [57,64]. Ketones, epoxides, esters, nitriles, alkenes, and alkynes are not affected by this reagent.

## 4. Multicomponent Ugi Reaction

Multicomponent reactions attract considerable attention because of their ability to synthesize complex structures in a few steps. A four-component Ugi reaction is a common multicomponent system. The amine, carbonyl compound, isocyanide, and carboxylic acid react together to give bis-amides [65]. The “separation” of the primary amine into two secondary amines makes it possible to expand the skeleton of the product of the multicomponent reaction (*N*-split Ugi) (Figure 19). One of the two nitrogen atoms of diamine reacts with the carbonyl compound to give an imine ion (in place of the protonated imine), which then reacts with the isocyanide and the carboxylic acid. As a result, the intermediate undergoes a transacylation, in which the acyl fragment moves to the second nitrogen atom of the diamine (a “remote” Mumm rearrangement) to give the α-acylaminoamide, whereas the first nitrogen atom has already become a tertiary amine, unable to obtain an acyl group [66].

The synthesis of PA **73** based on the *N*-split Ugi reaction was developed in [55], involving the interaction of an *N*-acetylamino acid **71**, *N*,*N*′-dibenzyl protected diamine **68**, paraformaldehyde (**69**), and isocyanide **70** (Figure 20). Together with α-acylaminoamide **72a**, aminal **72b** was formed, which hindered the formation of the target product and reduced the yield of the target compounds. The maximum yield was obtained for the amide with the three-methylene spacer group, owing to its conformational mobility. The reduction of amide groups was performed using BH_3_·THF. Benzyl groups were removed using Pd(OH)_2_/C. The highest yields were observed; then, cyclic diamines (piperazine and homopiperazine) were used.

For the preparation of new aminoamides, a new method that utilizes the competitive interaction of iminoanhydride with the Mumm rearrangement was developed [67]. In this case, 2-(hydroxymethyl)benzoic acid first reacted in a classical *N*-split Ugi reaction to give the intermediate, which underwent intramolecular cyclization because of the acidity of the aromatic hydroxyl group, resulting in the formation of phthalide and β-aminoamide (Figure 21). The use of two equivalents of carbonyl compound, isocyanide and 2-(hydroxymethyl)benzoic acid, allowed symmetric bis-(β-aminoamide) to be obtained in one step.

## 5. Synthesis of Polyamines for Clinical Trials

In cancer cells, the concentration of PAs increases dramatically. This discovery led to the creation of synthetic analogs of PAs as new antiproliferative agents. The common databases have reported numerous synthetic procedures for the preparation of different PAs and their conjugates. Some of these are in clinical trials.

### 5.1. AMXT-1501

The lysine-spermine conjugate (AMXT-1501) was found to be a potent PA-transport blocker when combined with difluoromethylornithine (DFMO). DFMO multiplies the effect of AMXT-1501, and their combination significantly decreased both the intracellular concentration of PAs and tumor cell growth [68].

Spermine was coupled with an l- or d-stereoisomer of the orthogonally protected ester Boc-Lys(Cbz)-ONp (**74**) to give mono- and diacylated products (Figure 22). After Boc_2_O treatment, compound **75** was isolated and subjected to catalytic hydrogenation to remove the Cbz protecting group. The terminal amino group was then functionalized with palmitic acid. Boc groups were removed under acidic conditions to yield AMXT-1501 [69].

### 5.2. SAM486A

SAM486A is the most potent AdoMetDC inhibitor, which was synthesized starting from 4-aminoindanone **76** (Figure 23). Compound **76** was converted to compound **78** via thioamide **77**, which was *S*-alkylated with Meerwein’s reagent. The intermediate **78** was treated with aminoguanidine to form the crystalline non-hygroscopic dihydrochloride salt **79 [70]**.

SAM486A inhibited non-small lung cancer cell growth (EC_50_ = 0.286 µM and 0.906 µM for A549 and H1299, respectively) [71]. The second phase of clinical trials showed the lack of beneficial therapeutic activity of SAM486A in patients with metastatic melanoma. The use of this agent led to increased fatigue, a nascence of myalgia, and a decrease in neutrophil concentration. Biopsy results confirmed that SAM486A inhibits AdoMetDC, promoting an increase in putrescine level and decreasing spermidine concentration [72].

## 6. Synthesis of Polyamines Conjugates and Their Biological Activity

### 6.1. Conjugates with Porphyrins (Chlorin e_6_)

The synthesis of the PA–Chlorin *e*_6_ conjugates **82a**,**b** required the selective protection of the nitrogen atom of spermidine and spermine [73,74]. The starting Chlorin *e*_6_ **80** was treated with regioselectively Boc-protected PAs **81a**,**b**. The conjugates obtained were subtracted to acidic hydrolysis of the Boc-protecting groups to form desired conjugates **82a**,**b** (Figure 24).

The photocytotoxicity of compounds **82a**,**b** against a human chronic myelogenous leukemia cell line (K562) was evaluated and compared with Photofrin II^®^ and Chlorin *e*_6_. PA–Chlorin *e*_6_ conjugates caused significant cell death either immediately after irradiation or after 24 h in the dark. Spermine and spermidine conjugates showed the same activity, which decreased after 24 h of incubation. Chlorin *e*_6_ **80** exhibited low photocytotoxicity, while Photofrin II^®^ possessed weak activity.

### 6.2. Conjugates with Boron Cluster

Boron neutron capture therapy (^10^B-NCT) is a promising tool in the treatment of malignant tumors, primarily intractable brain tumors. This method is used to treat predominantly progressive or metastatic tumors in cases where the possibilities of adjuvant therapy have been exhausted. Boron clusters [B_n_H_n_]^2−^ (*n* = 10, 12) are known to form an extensive class of closed inorganic systems characterized by thermal stability, resistance to oxidants, and the ability to replace exopolyhedral hydrogen atoms with various functional groups. Derivatives of this type react with neutral and negatively charged nucleophiles to form compounds with a functional group separated from the cluster by alkoxy spacer (side functional group). Derivatives of *closo*-borate anions with oxonium-type substituents can act as starting compounds for further modification, owing to opening reactions of cyclic substituents, which makes it possible to introduce biologically active groups into the boron cluster [75,76,77].

The treatment of compound **83** with hydrazine or PAs was shown to proceed as the oxonium ring opening followed by the addition of a side terminal amino group, yielding the PA conjugates **84a**–**f** (Figure 25).

The corresponding ring-opening reaction of the monosubstituted *closo*-decaborate anion, intended for other PAs, opens up a new pathway for further modifications of boron clusters [78].

### 6.3. Conjugates with Flavonoids

A simple approach to the synthesis of anticancer PA–flavone–naphthalimide conjugates was described in [79]. The condensation of **85** with the PAs and the acidic hydrolysis of Boc groups with 4M HCl led to the desired compounds **86a**–**g** in 65% yields over two stages (Figure 26).

Biological activity in vitro was performed to assess the inhibitory effect of the conjugates **86a**–**g** on tumor cell lines SMMC-7721 and HepG2 and on normal hepatocyte QSG-7701 [79]. Compounds **86a**–**c** with linear mono-, di-, and tri-amine residues exhibited better activity in the MTT test than those with branched (**86d**) or cyclic (**86f**,**g**) amines. The terminal amino group of PA–flavone–naphthalimide conjugates played an important role in their inhibitory properties. Additionally, the high antitumor activity of the homospermidine-based compound **86c** was determined for the cell lines SMMC-7721 and HepG2, whereas the IC_50_ value for normal hepatocyte was greater than 50 μM.

The PA–flavonoid and PA–naphthalene–diimide conjugates were found to effectively inhibit tumor metastasis [80,81]. Compound **86c** markedly attenuated the migration of HepG2 and SMMC-7721 cells in a dose-dependent manner. In vivo studies showed that its inhibitory effect was higher than that of the commercial drug amonafide.

### 6.4. Conjugates with Artesunate and Trioxolane

Conjugates of PAs with artesunate and trioxolane were found to exhibit varying levels of antiplasmodic activity [82]. Symmetric conjugates **89a–d** were synthesized by the activated esters method in 65% yields, except for compound **89c** (37%) (Figure 27). Compounds **89a**,**b** were screened against the *P. falciparum* K1 strain, which has chloroquine resistance. To determine the selectivity index, the cytotoxicity of the compounds was also evaluated in relation to the myoblasts and the rat skeleton L6 cells.

Surprisingly, strong antimalarial activity in vitro for conjugates **89a** and **89c** (IC_50_ activity values ranged from 0.3 to 1.1 nM) was noted [82]. A study of the antimalarial activity of conjugates showed that an increase in cytotoxicity was paralleled with the chain length increase between internal nitrogen atoms. Conjugates of PAs with artesunate **89b,e** had better antimalarial properties than the corresponding analogs with trioxolane **89a**,**c** and the starting trioxolane acid **88a**.

### 6.5. Conjugates with Acridine

For the synthesis of PA–acridine conjugates **92a**–**c** and **96a**–**c**, which exhibit antimalarial activity, a synthetic strategy was developed according to the Gabriel method. Amines **90a**–**c** were treated (Figure 28) with commercial 6,9-dichloro-2-methoxyacridine (**91a**) or 9-chloroacridine (**92b**); after the removal of the protective groups, the compounds were obtained in a total yield of 30–68% [83].

The new acridine derivatives exhibited different antimalarial activities that depended on acridine substituents, side-chain length, and the nature of the main chains. In the series of aliphatic diamine (**95a**–**c** and **96a**–**c**) and piperazine (**92a**–**c** and **93a**–**c**) derivatives, the antimalarial activity decreases with the increasing methylene group numbers. In the case of morpholine derivatives **94a**–**c**, the length augmentation improved the activity and maintained its invariably.

### 6.6. Conjugates with Ferrocene

The inclusion of transition metals in a drug structure often enhances its activity. Neuse’s work [54] involved a group of nontoxic and stable organometallic derivatives of *bis*-(g5-cyclopentadienyl)iron (II))-ferrocenyl (Fc). The inclusion of the Fc group opens new perspectives for therapeutic applications and altered drug resistance. Schiff bases were shown to have usability and efficiency in the synthesis and modification of PAs with Fc substituents (Figure 29). Aldehydes were added to the starting ferrocenyl diamine **97**, followed by the reduction with NaBH_4_ to obtain PA–Fc conjugates **98a**–**h** [53].

The compounds were tested against *T. brucei* and *T. cruzi* strains using the MTT assay to evaluate trypanocidal activity. Only two compounds **98d** and **98h** showed low activity against the *T. brucei* strain. For the *T. cruzi* strain, better activity was found for compound **98f** with *o*-methoxyphenyl residue.

### 6.7. Lipophilic Polyamines

Regioselectively protected PAs have also been used in the synthesis of lipophilic derivatives as non-viral liposomal gene delivery systems. The polycationic nature of lipophilic PAs significantly enhances the condensation and protection of nucleic acids (NAs) in the liposome/NA complexes, enabling the delivery of NA into various cells to change or correct their functions [84].

Nucleolipids have attracted attention in medicinal chemistry as molecular devices and various therapeutic agents [85,86]. They can bind NAs through hydrogen bonding, π–π stacking, and electrostatic interactions. They were found to be less toxic than the commercial agent Lipofectamine 2000. Therefore, to improve siRNA (small interfering RNA) folding and delivery into cells, PAs-based nucleolipids **104a**–**c** were synthesized (Figure 30).

In the first step, the 5′-OH group of uridine **99** was protected by the DMTr group. Hydrophobic oleyl residues were introduced to the product to give a lipophilic uridine derivative **101** in 60% yield. After the removal of the protecting DMTr group under acidic conditions, compounds **102** were coupled with various PAs in the presence of CDI and DMAP to form a uridine derivative. Amino groups were protected in situ with Boc_2_O, and compounds **103a**–**c** were easily isolated on silica gel in 42–70% yields. Final deprotection led to the formation of lipophilic PAs **104a**–**c** in the form of hydrochlorides in quantitative yield [87].

The siRNA concentration, liposome-to-siRNA ratio, and the PA structure were the main factors that influenced transfection efficiency. Complete siRNA binding was afforded at the minimal liposome/siRNA ratio of three. The liposome sizes varied from 20 to 200 nm, and their surfaces were positively charged. In addition, liposomes did not affect cell viability, and they successfully delivered siRNA into cells in vitro, causing the suppression of the target mRNA at an N/P ratio of 12.

A typical structure of gemini-amphiphiles applied in gene delivery systems consists of PAs (spermine or triethylenetetramine) linked with two cholesterol residues via spacer groups of various types. The synthetic strategy (Figure 31) implied condensation under the Fukuyama conditions of regioselectively protected PAs **105a**,**b** with bromo derivatives of cholesterol **106a**–**c** and the subsequent deprotection, which yielded PA-cholesterol gemini-amphiphiles **107a**–**c** [88,89].

A more complex approach (Figure 32) was used for the preparation of disulfide PA-cholesterol amphiphiles [90,91,92]. Here, dicarboxy derivatives **108a**,**b** were obtained from regioselectively protected spermine **105b** by coupling with bromobutanoic acid, followed by deprotection and the hydrolysis of the resulting ester. Then, the dicarboxy derivatives **108a**,**b** were condensed with the amino derivatives of cholesterol **109a**,**b** in the presence of EEDQ. The Boc deprotection resulted in lipophilic polyamines **110a**,**b**.

The cationic liposomes formed from PA-cholesterol amphiphiles **107a**–**c** or **110a**,**b**, and the zwitterionic lipid DOPE delivered DNA or siRNA into HEK293 and BHK IR-780 cells. Thus, compound **107c** with a hydrophilic ethoxyethoxyethyl spacer was the best transfectant for DNA, whereas siRNA transfer into the cells was efficiently mediated by liposomes consisting of lipophilic polyamines **107b** with a hydrophobic octamethylene spacer or disulfide compound **110a**.

Asymmetric lipophilic PAs were found to be promising antitumor agents. A number of amphiphiles based on triethyltetramine **111a**,**b**, norspermine **112a**,**b**, and spermine **113a**–**f** were synthesized using the Fukuyama reaction [93]. Compounds **111**–**113** had various long-chain alkyl substituents at the C(1) glycerol atom and PA residues, which could be alkylated at the terminal amino group (Figure 4).

Lipophilic PAs were tested for antitumor activity against tumor cell lines of adenocarcinoma of human chronic myelogenous leukemia (K562), colon (HCT116), and mammary glands (MCF7) using an MTT assay. All compounds affected tumor cell death at micromolar concentrations, and they were four to five times more potent than the reference anticancer agents: Edelfosine [94,95], DENSpm [21], and cationic lipid (*rac*-*N*-methyl-*N*′-{4-[(2-ethoxy-3-octadecyloxy)prop-1-yloxycarbonyl]butyl}imidazolium iodide) [96]. The highest cytotoxicity was observed for triethyltetramine-based lipophilic PAs **111a**,**b** or norspermine-based lipophilic PAs **112a**,**b**. For all of the cell lines tested, the cytotoxicity of the spermine-based compounds **113a**–**e** was hardly influenced by the length of the *O*-alkyl chain at the C(1) position of glycerol backbone.

### 6.8. Conjugates with Hydroxycinnamic Acids

The synthesis of antifungal PA–hydroxycinnamic conjugates was fully described by Kyselka [97]. The initial derivatives of hydroxycinnamic acid **114a**,**b** were treated with Boc-putrescine **115**, followed by the removal of protective groups and the formation of conjugates **116a**,**b** (Figure 33). For the synthesis of di- and tri-substituted coumaroyl spermidines, acetoxycinnamoyl chloride (**117**) was conjugated with unprotected spermidine to form the desired products **118a**,**b**.

PA–hydroxycinnamic conjugates had various effects on the development and survival of fungi. Compound **116a** had the greatest effect, inhibiting the growth of various mycooxygenic fungi at 2.5–10 mM, whereas compound **116b** was less active. The presence of the methoxy group reduced the antifungal activity of conjugates, owing to a slight increase in the compounds’ polarity, which decreased their interactions with the outer membrane of the fungi.

## 7. Conclusions

PAs are directly involved in the development of a wide range of pathogenic pathways. Synthetic analogs of PAs and their conjugates have been shown to be promising candidates for the treatment of many diseases, including myelogenous leukemia; hepatocarcinoma; and colon, breast, and lung cancer. The synthesis of PAs is a complicated task because of their high polarity, the presence of several amino groups, and multistep procedures necessary for their preparation. Currently, most efforts are focused on the development of inexpensive, mild, effective, and scalable synthetic approaches. The use of the multicomponent Ugi reaction and its modifications allows researchers to obtain the PAs of different structures over two steps in high yields from simple starting precursors.

The investigations of biological activities of PA conjugates mentioned in this review show their broad therapeutic applications and clinical potential. They are promising solutions for the development of NA delivery systems, antitumor agents for photodynamic and boron neutron capture therapies, and antimalarial and antitrypanocidal compounds.

## Data Availability

Not applicable.

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
