# Peer review of "Recent Advances in the Synthesis of Polyamine Derivatives and Their Applications"

_molecules, 2021, doi:10.3390/molecules26216579_

Round 1

Reviewer 1 Report

The manuscript presented Nichugovskiy et al, reviewed the methods for polyamines synthesis. Different routes were presented: 1- syntesis (Alkylation, Acylation, Imine formation, Michael reaction, Fukuyama amine synthesis, Solid phase synthesis, Regioselective protection of amino groups, Cleavage of the nitrogen-containing cycle, Opening of the oxygen-containing cycle) 2- Reduction, Multicomponent Ugi reaction. After this general review, the authors presented important polyamines that are currently in clinical trials, for application in tumor and cancer therapy.

The synthesis mechanisms were presented and in my opinion are important to illustrate and differentiate the different synthesis routes.

In my opinion the review manuscript is very complete and reports recent and important advances in polyamines synthesis.

In order to improve the quality of the manuscript, I recommend the authors to improve the abstract section:

1-In the Abstract, the authors focused on the in vivo application of polyamines. They should include the main synthesis routes reviewed in the manuscript together with the polyamines that are in clinical trials.

In my opinion the manuscript could be accepted after minor revision.

Author Response

Dear Reviewer,

According to your comments we revised our manuscript.

In order to improve the quality of the manuscript, I recommend the authors to improve the abstract section: In the Abstract, the authors focused on the in vivo application of polyamines. They should include the main synthesis routes reviewed in the manuscript together with the polyamines that are in clinical trials.

Thank you very much for all your remarks. We have rewritten abstract taking into account your comment.

Abstract: Biogenic polyamines (PAs) are involved in the growth and development of normal cells, and their intracellular concentration is stable. The concentration of PAs in cancer cells is significantly increased to promote and sustain their rapid proliferation. Over the years, synthetic PAs, which differ in their structure, have demonstrated high antitumor activity and are involved in clinical trials. The chemical synthesis of PAs and their conjugates require the correct choice of synthetic pathways—methods for constructing conjugates and the orthogonal protection of amino groups. The most common methods of synthesis of PA conjugates are acylation of regioselectively protected PAs or their alkylation under the conditions of the Fukuyama reaction. One of the most promising method of PAs synthesis is the use of multicomponent Ugi reaction, which allows various PAs to be obtained in high yields. In this review, we described and analyzed various approaches that were used in the synthesis of polyamines and their conjugates.

Sincerely,

Dr. Mikhail Maslov

Reviewer 2 Report

Presented work is well written and interesting, so it can be accepted after minor revisions.

Lines 33-36: Paragraph can be completed by more informations about PAs role in cell apoptosis (including tumor cells). Some anti-tumor strategies are related to catching of PAs in cells (for example: https://doi.org/10.1038/s41467-019-11553-7).

Introduction should also contain some general informations (references) about the PAs and cancer (for example: https://doi.org/10.1007/978-1-4939-7398-9_39; https://doi.org/10.1016/j.canlet.2021.06.020).

In my opinion introduction can be also completed by short information about other applications of PAs (like putrescine, spermine and spemidine) and their derivaites, which are connected with antifungal and antimcrobial applications (for example: https://doi.org/10.1021/acs.jafc.8b03976, https://doi.org/10.1016/j.foodhyd.2018.08.008). This suggestion is completly optional (decision belongs to the Authors).

Author Response

Response to Reviewer

Dear Reviewer,

According to your comments, we revised our manuscript.

Lines 33-36: Paragraph can be completed by more information about PAs role in cell apoptosis (including tumor cells). Some anti-tumor strategies are related to catching of PAs in cells (for example: https://doi.org/10.1038/s41467-019-11553-7).

Thank you for the link to an interesting article, we read it and added some information in paper.

«Of note, the lack of PAs in cells leads to the activation of apoptosis; therefore, a prom-ising anticancer target is the inhibition of PA synthesis [8,9]. Additionally, Chen’s work [10] describes the mechanisms of trapping natural PAs in cells because of their binding to peptide-pillar[5] arene conjugates, which show high antitumor activity against breast cancer.»

Introduction should also contain some general information (references) about the PAs and cancer (for example: https://doi.org/10.1007/978-1-4939-7398-9_39; https://doi.org/10.1016/j.canlet.2021.06.020).

We have added material about the transport system of polyamines and its effect on cancer cells.

«PAs can be trapped from exogenous sources. The polyamine transport system (PTS) is known to involve the absorption and withdrawal of PAs, which are necessary for their processing. The molecular structure of the mammalian PTS still remains, but the most preferred PA transport is linked to glypican-mediated endocytosis, pinocytosis, and caveolin-mediated endocytosis [12,13]. PTS is not highly specific and can transfer other molecules (Figure 3), such as paraquat, methylglyoxal bis(guanylhydrazon) (MGBG), mepacrine, and polyamine-based compounds. The amount of transported compound increases when the delivering molecule resembles the structure of spermidine or spermine. Also, the data in the literature indicate that the primary amino groups in the PA structure are not sufficient for their uptake [14]. PTS can be considered as a target for cytotoxic polyamine-conjugated drugs [15].»

In my opinion introduction can be also completed by short information about other applications of PAs (like putrescine, spermine and spemidine) and their derivaites, which are connected with antifungal and antimcrobial applications (for example: https://doi.org/10.1021/acs.jafc.8b03976, https://doi.org/10.1016/j.foodhyd.2018.08.008). This suggestion is completly optional (decision belongs to the Authors).

Since our article is of a chemical-synthetic nature, we decided to add a wonderful article about antifungal conjugates of polyamines (paragraph 6.8). The second article is not suitable for our review article, so we did not include it.

«6.8. Conjugates with hydroxycinnamic acids

The synthesis of antifungal PA-hydroxycinnamic conjugates was fully described by Kyselka [100]. The initial derivatives of hydroxycinnamic acid 114a,b were treated with Boс-putrescine 115, followed by the removal of protective groups and the formation of conjugates 116a,b. For the synthesis of di- and tri-substituted coumaroyl spermidines, acetoxycinnamoyl chloride (117) was conjugated with unprotected spermidine to form the desired products 118a,b.

Scheme 33. Synthesis of antifungal PA-hydroxycinnamic conjugates. Reagents and conditions: a) DCC, DCM; b) TFA, DCM; c) HCl/MeOH; d) TEA, THF; e) NH3/MeOH.

PA-hydroxycinnamic conjugates had various effects on the development and surviv-al of fungi. Compound 116a had the greatest effect, inhibiting the growth of various my-cooxygenic fungi at 2.5–10 mM, whereas compound 116b was less active. The presence of the methoxy group reduced the antifungal activity of conjugates, owing to a slight increase in the compounds’ polarity, which decreased their interactions with the outer membrane of the fungi.»

Thank you very much for all your remarks.

Sincerely,

Dr. Mikhail Maslov
